# A University’s Role in Developing a Regional Network of Dementia Friendly Communities

**DOI:** 10.3390/ijerph22050721

**Published:** 2025-05-01

**Authors:** Laurel Standiford Reyes, M. C. Ehlman, Suzanne Leahy, Reagan Lawrence

**Affiliations:** 1Human and Family Development Lab, Psychology Department, University of Southern Indiana, 8600 University Boulevard, Evansville, IN 47712, USA; 2Bronstein Center for Healthy Aging and Wellness, Kinney College of Nursing and Health Professions, University of Southern Indiana, 8600 University Boulevard, Evansville, IN 47712, USA; mehlman@usi.edu (M.C.E.); sleahy@usi.edu (S.L.); rwlawrence@eagles.usi.edu (R.L.)

**Keywords:** dementia, community-based supports and service, caregiver support, Dementia Friendly America

## Abstract

Introduction: The World Health Organization has identified dementia as a growing global health concern with 10 million new cases diagnosed every year. The growing number of people living with dementia (PLWD) heightens the need for effective interventions that support PLWD and their caregivers. The most effective interventions supporting PLWD and caregivers combine education, care, and services to increase knowledge, decrease stigma, improve care, heighten empathy, and increase engagement of PLWD in their communities. Dementia Friendly America (DFA), administered by USAging, promotes a Dementia Friendly Community (DFC) initiative designed to engage multiple sectors (e.g., business, healthcare, community services) and engage PLWD in a comprehensive community change process. A center for healthy aging and wellness at a midwestern public university developed a network approach in its regional support of eight DFCs, as a part of its Geriatric Workforce Enhancement Program funded by the U.S. Health Resources and Services Administration. Objective: This article documents a mid-size university’s approach to establishing a regional DFC network of urban and rural communities surrounding the university, describing the support the university provided as well as how communities implemented the four-phase DFC process and emulated guiding principles. Results: A retrospective evaluation found engagement with the DFA guiding principles and varying levels of adherence to DFC phases. Discussion: The project team suggests that there are unique roles that universities can play in supporting the DFC movement and that developing a network of communities is a helpful strategy to use in providing this support. Additionally, the authors propose the integration of a community change model to guide future DFC work. Conclusions: This article helps to fill an existing research gap concerning DFC implementation and explores the unique role academic partners can play in cultivating regional hubs of DFC activity.

## 1. Introduction

On 1 October 2024, the World Health Organization (WHO) called for “urgent transformation of care and support systems for older people” [1] in response to the UN findings that older people experience “long-standing systemic inequalities and challenges to human rights deeply rooted in ageism and reflected in inadequate health services for older people, gaps in social protection and intersecting discrimination based on age, gender, disability, and other grounds” [2] (p. 10). To combat these challenges, it is important to fight stigmas that fuel these disparities, including the highly stigmatized disease of dementia, which disproportionately affects older people. Research has suggested that interventions that increase awareness and understanding of the aging processes [3], provide intergenerational and multi-ability community events and resources [4], and recognize the contributions and value of older people in society [5] are needed to transform the care and community supports available to older people. Addressing the misinformation and stigmatization surrounding aging and diseases, such as dementia, is of utmost importance [6].

### 1.1. Dementia Care and Stigma Concerns

Dementia is a growing global health issue, with 10 million new cases diagnosed annually [7,8] despite barriers to detection and diagnosis, including stigma [9], access to care [10], and frequent misdiagnoses [11] or underdiagnosis [12]. Recent estimates by Dhana et al. [13] indicate that dementia impacts more than 10% of individuals aged 65 and older in most U.S. states and exceeds 33% of people over 85 years. Currently, approximately 7 million Americans are living with Alzheimer’s Disease alone, contributing to an annual cost exceeding $300 billion [13].

The stigmatization of people living with dementia (PLWD) makes this chronic disease particularly challenging to address [14]. People diagnosed with dementia are stigmatized, isolated, excluded, and made invisible, reflecting pervasive misinformation about the disease. Social isolation and loneliness, in turn, exacerbate the condition and contribute to additional physical and mental health conditions [7,15,16]. The WHO emphasizes that the remedy lies in fostering meaningful connections [17,18].

Dementia knowledge and a personhood focus—in contrast to a disease focus—decrease fear and stigma and increase the likelihood of connection by emphasizing and valuing the individuality, preferences, and capabilities of each person [14,19]. Stigma is decreased most effectively through educational efforts and meaningful intergenerational activities that highlight the abilities, experiences, and personhood of PLWD [6,11,14,20]. As caregivers and providers of PLWD are increasing efforts to move from disease-focused care to person-focused care, support and awareness from the whole community becomes increasingly important.

### 1.2. The Importance of Community Engagement

Individuals living with dementia and their caregivers face complex needs that hinder their ability to maintain community engagement [11,21,22,23]. The absence of community engagement is a risk factor for social isolation and loneliness, defined objectively as having few social groups and social interactions [24]. Social isolation and loneliness expedite the onset of dementia [7]. With the majority of PLWD cared for by family caregivers [23,25] in the community, an emphasis on increasing community support and resources is needed.

Communities may fall short of providing the necessary support for PLWD and their caregivers; services also may be unused due to the stigmatization of the disease [11,23]. Wiersma and Denton [23] discovered that many influences and barriers determine “family caregivers’ use of community-based services, including stigma, the lack of privacy, beliefs and attitudes, lack of awareness of services, financial barriers, acceptability of services, and challenges in service delivery” (p. 15). To address these, several global initiatives aimed at increasing community engagement appear promising and have yielded positive outcomes [20,26,27], particularly those that emphasize incorporating the decision-making voice of PLWD into these initiatives, while creating space to share their experiences, strengths, and talents [6,28]. Communities that focus on a combination of education, care, and support for PLWD and their caregivers show decreases in stigma, improvements in care, heightened empathy, and increased engagement of caregivers and PLWD [11,20,29].

#### 1.2.1. Dementia Friendly America

The Dementia Friendly America (DFA) movement, founded in 2015, has emerged as an effective model for implementing dementia-friendly initiatives [11]. DFA aims to “raise dementia awareness, reduce stigma, and provide high-quality dementia-friendly resources, education, and support” [30]. Its goal is to foster the creation of communities where PLWD and their caregivers can thrive [30]. DFA seeks to address social isolation by offering safe environments for individuals with dementia, their caregivers, and families, while maintaining an informed, secure, and respectful approach to those affected [31]. A dementia-friendly community is informed, safe, and respectful of individuals with the disease, as well as their families and caregivers, and offers supportive options that foster quality of life. By becoming a DFC, a community commits to becoming more dementia-friendly [30].

DFA challenges communities to respond to dementia with a focus on cultural, structural, and environmental changes. The DFA Community Toolkit (n.d.) was developed by DFA to support communities in becoming more dementia-friendly. The toolkit defines a four-phase process, aligned with research-based models of community change, highlighting the importance of education, the assessment of supports and barriers for PLWD and their caregivers, the use of assessment results in community goals and planning, and the consistent involvement of PLWD in all stages of assessment, planning, and programs [26]. The four phases address community readiness (Convene), data collection (Engage), analysis and prioritization (Analyze), and action plan development (Act). On its website, DFA also identifies two guiding principles of DFCs: Principle 1: Include and involve PLWD in the community effort and Principle 2: Establish and maintain a team that works collaboratively to create change.

Although there is limited research about the specific implementation processes of communities [26], positive outcomes are associated with DFCs. Some examples include community cohesiveness and the dissemination of educational opportunities to community members [21,23], as well as the social engagement of individuals living with dementia and their caregivers [11,29,32]. Community cohesiveness allows for meaningful connections that serve as a “safety net” for PLWD and their caregivers [23]. Education is key to building a safety net as it equips members with knowledge and practical ways to support those affected [11]. Phillipson et al. (2019) also found that educational programs reduce stigma toward aging and dementia, fostering greater understanding and empathy [6].

#### 1.2.2. A University’s Role

Universities often combine efforts with grass-roots initiatives in their surrounding communities. This reciprocal collaboration can lead to sustainable changes in environments, where all parties have a sense of investment and “ownership” of the programs or initiatives [33]. For example, creative solutions to unique challenges are achieved in settings where diverse people from the community and academia work together towards a common goal, using the strengths and talents of everyone [34,35,36]. The additional positive outcomes of these collaborations include a lower incidence of the marginalization of groups, more effective communication, higher cooperation and trust, a better understanding of what can be accomplished, and more effective knowledge dissemination and application to real-world issues [37].

Many universities work with communities as a strategy for increasing the support of PLWD and their caregivers. Generally, universities seek to create a dementia-literate workforce by preparing future professionals in their disciplines [38,39,40]. Other efforts are specific community initiatives, such as a university offering an art program at a memory care center or art kits to have at home [41]. Universities also work to make the campus and events more generally dementia-friendly [42] or offer various research support [29] to show the effectiveness of community-established programs that support people affected by dementia and other resources [43]. Universities like Dementia Friendly Iowa [44] and Dementia Friendly Nevada [45] have recognized roles for universities in the dementia-friendly community space. The value of dementia-friendly initiatives is recognized. However, the published literature on university–community dementia initiatives is limited to the description of DFC implementation. No available research was found on the use of a network to support DFC implementation or on the unique role that academic settings can play in building regional hubs of DFC activity.

### 1.3. The Current Project/Purpose

Our university sought to collaborate with the surrounding communities to raise awareness of dementia and available resources while decreasing stigma. While DFCs offer a promising strategy for scaling dementia interventions and reducing stigma, community change can be slow and hard won [36]. As the university’s DFC program evolved, we developed a network strategy for sharing resources and the technical and research assistance that university staff could provide. Creating and testing programmatic models for developing dementia-friendly community networks is critical in response to the expected global and national growth of the population of individuals living with dementia and their caregivers. There also appears to be limited information about university efforts to support DFC creation. This article outlines a university’s approach to creating a network of eight DFCs (both urban and rural) in the region surrounding the university, documenting the support the university provided, as well as how each of the communities adhered to the community process and principles promoted by DFA. We also discuss the implication that universities can assume important roles in collaborative efforts to support community change.

## 2. Materials and Methods

A midsize state university was awarded a five-year Health Resources and Services Administration Geriatric Workforce Enhancement Project (GWEP) in 2019. One major objective of the project included establishing three Dementia Friendly Communities (DFCs). With already five years of statewide workforce development in building dementia knowledge and skills, the awardee was poised to provide training and support DFC infrastructure building in communities. In preparation, the university staff participated in workshops on the Dementia Friendly America (DFA) four-phase framework and two guiding principles for DFC building. Over the course of three years, eight communities joined what would become a regional network. Given the success of the network in attracting new communities, a retrospective evaluation was conducted to examine the collective accomplishments and regional opportunities for advancing regional DFC work. (Readers may find it helpful to consult the list of acronyms provided in Appendix A).

### 2.1. Network Development

What started as a plan to launch three DFCs, one per year, evolved into a network of eight DFCs in a 12-county region. Despite the challenges of the 2020 COVID-19 pandemic, individuals mobilized to create DFC communities for PLWD and caregivers. In collaboration with two regional Area Agency on Aging (AAA) partners, the university GWEP identified and invited one DFC per AAA service region to participate during the second year of the grant. This first cohort of DFCs was followed by three additional cohorts of communities, two each grant year, for a total of eight DFCs launched as a part of the university GWEP program. The last two cohorts were comprised of communities that were not recruited by the university; these communities sought the university GWEP DFC program to join.

#### Description of Participating Communities

Table 1 shows the four cohorts of communities that took part in the GWEP DFC program. A new cohort was introduced to the GWEP annually and consisted of two new communities interested in becoming a Dementia Friendly Community. Participating DFCs ranged in geographic scope from towns, neighborhoods (sub-city) and counties. Each of the eight DFCs represented a different county of the midwestern state, except for communities 2B and 3A, which were separately initiated on different sides of the same city.

The university supported the development of a network among DFC communities by providing technical support in the form of both administrative resources and community-level resources. Administratively, the university dedicated staff support and leveraged relationships with the AAA. In the first two years of the project, the university GWEP program coordinator was able to support the DFC efforts at approximately 0.1 FTE. For the final three years, this allocation increased to approximately 0.2 FTE. The program coordinator supported stakeholder information meetings, completed paperwork for DFA recognition, and assisted with aspects of the DFC change process. Additionally, undergraduate and graduate assistant time was allocated to support the DFA community network at various periods of the project. A key feature of the university-based GWEP DFC network was establishing an Area Agencies on Aging partner to act as an action team liaison (facilitator), as well as a champion for the other DFCs. One care manager was funded at 0.05 FTE to support the DFC work.

Community-level resources were allocated to fund materials and supplies, and to support the action-team infrastructure. Each DFC received funding for materials and supplies supporting initiative implementation. As such, more than half of the DFCs utilized funding to certify an action team member in Dementia Live^®^, a training program designed to build knowledge and empathy by simulating the experience of living with dementia. DFCs partnered with local libraries to circulate Memory Kits, backpacks of activities designed to create conversations and strengthen relationships between people living with brain change and their caregivers. Each DFC also was provided a collection of supportive supplies such as robotic pets and fidget blankets as community resources. Finally, to complement the online DFA resources, the university used a Microsoft (MS) Teams site as a communications platform for the regional DFC network to house action team sample agendas, PowerPoints, and other resources. The university also met with other state AAAs to learn about Dementia Friends initiatives (another initiative of Dementia Friendly America) and, subsequently, joined the newly established Dementia Friendly Indiana. Two university GWEP staff attended a national conference on DFC development as well. Table 2 outlines the DFC selected activities and university technical support provided by the DFA four-phase model.

With sustainability in mind, at the conclusion of the grant the university network created a DFC Resource Corner website as a complement to the national DFA website. The university network hosted a DFC meeting marking the end of the five-year GWEP funding to celebrate the DFCs successes, share participation data and initial findings from the regional needs assessment, and to unveil the DFC Resource Corner website. Staff presented evaluation results to DFC action teams attending the grant-end celebration and engaged teams in how these results might inform community plans. Program evaluation was not an area of technical assistance that the university undertook due to the emphasis on completing the four-phase process as well as grant resource limitations. The DFC Resource Corner, however, does include some evaluation resources to supplement the Dementia Friendly Community Evaluation Guide available on the DFA website.

### 2.2. Retrospective Evaluation Methods

The university hoped to learn more about the network successes and provide communities with information that could be used to strengthen the regional DFC work. The evaluation examined what DFC implementation looked like in the region and the extent to which implementation had adhered to the four-phase community development model outlined by Dementia Friendly America on its website, the online Community Toolkit, and additional online resources. The evaluation also examined the extent to which the two guiding principles (1. Involve PLWD in the community effort; and 2. Maintain a team that works collaboratively to create change) underpinning the DFA model were exhibited by regional DFCs. Finally, the evaluation examined the alignment between regional DFC efforts and the work of other DFCs nationally, highlighted in the Dementia Friendly Community Evaluation Guide [47].

The evaluation, conducted by the dedicated GWEP evaluator, involved the secondary analysis of data collected during the initiative. There were two primary sources of evaluation data: (1) DFC coordinators were asked to report activities monthly using a Qualtrics-based reporting form (see Appendix A); and (2) a roster of Action Team members who collected information about community sector representation (see Appendix A). In addition to collecting DFC activities, the Qualtrics form asked DFC coordinators to report on any new members joining the Action Team and nudged coordinators to enter contact information into the Action Team roster, stored on a shared MS Teams channel to facilitate real-time updates. Quarterly, university student assistants updated the individual DFC rosters with any new names provided in the Qualtrics form that were not added by the DFC coordinator. A chronological list of DFC events and each DFC’s Action Team roster were verified on a semi-annual basis with individual DFC coordinators and/or the university DFC project coordinator.

A workflow was set up in Qualtrics to email coordinators a monthly reminder to complete the electronic reporting form. The monthly activity form asked coordinators to categorize each DFC activity as an Action Team meeting, community assessment, community education, use of Dementia Live^®^, community outreach, policy change work, support group, sector training, or other. For meetings and events, the electronic form collected the number of people in attendance, the length and format of the event, and the topical focus to report to the funder. The Qualtrics form also prompted coordinators to report on the number of participants who were informal caregivers, professional caregivers, or PLWD. Data were audited quarterly for completeness. Some coordinators sent copies of the meeting notes or event sign-in sheets in lieu of completing the monthly form on MS Teams. The university team recorded these data during quarterly audits.

The data analysis focused on answering three questions: (1) Did regional DFCs complete each phase of DFC development as described by the Dementia Friendly America model? (2) To what extent did regional DFCs exhibit the guiding principles of: i. Including and involving people living with dementia in the community effort; and ii. Establishing and maintaining a team that works collaboratively to create change? (3) Did the regional DFC work align with the goals of other DFCs nationally? The analysts applied rubrics to address these questions. The first rubric, Appendix A, contained each activity specified on DFA’s website for each of the four phases. Using the rubric, the analysts indicated whether there was no evidence of the activity, evidence that the activity had been partially implemented, or evidence that significant activity had been implemented for each of the communities in the region. Each phase was equally weighted in the scoring. A second rubric was created for the DFC principles (see Appendix A) using a similar scale of no, partial, or significant activity that evidenced the principle being scored. Finally, to answer the third study question, the analysts created a third rubric consisting of the six goals that were identified in the Dementia Friendly Community Evaluation Guide to determine whether these same goals fit the region’s activities and what might be learned from any patterns. The example activities listed in the Dementia Friendly Community Evaluation Guide for each goal were used to determine the placement of the activity on the rubric. Then, the analytic results from each of the rubrics were reviewed with the GWEP Project Coordinator and shared at the regional DFC convening for feedback and discussion. Additionally, one of the DFC coordinators was invited to review and comment on this article.

## 3. Results

### 3.1. Adherence to the DFC Community Change Process

#### 3.1.1. Phase One: Convene

There were four activities that the DFA identified as being important to carrying out phase one: (1) Convene key community leaders, citizens, and PLWD to determine community readiness to embark, evaluate, and sustain a DFC initiative; (2) Build the community case for becoming more dementia-friendly; (3) Build an Action Team consisting of key collaborators, PLWD, family and care partners, and community members representing different sectors; and (4) Engage the community more broadly by kicking off the effort in a community meeting or event [47]. The evaluation found that the focus of regional DFC activity fell into the latter two categories, building an Action Team and engaging the community. Across communities, there was evidence of fuller implementation of these activities than those focusing on assessing community readiness and building a community case. Appendix A provides results from the individual scoring of communities on each of the four activities.

The evaluation observed that none of the communities conducted a systematic community readiness assessment, despite the availability of validated tools to support this kind of work [48]. There was perhaps no need for a formal assessment, as DFC Action Teams reported that community leaders and families had insufficient knowledge on the disease and available resources. The lack of shared awareness and knowledge reported across the region led the university to adopt two strategies: 1. Support communities in assessing their informational and knowledge needs (discussed further under phase 2); and 2. Partner with the local Alzheimer’s Association chapters to help meet these needs.

Gaps in community adherence to phase 1 were not due to inattention or rushing forward to other subsequent phases of DFC implementation. Phase 1 represented an area of ongoing work for DFCs regardless of how long ago the DFC had been established. It may be that none of the DFCs reached the necessary threshold (e.g., five years) at which the phase 1 activities would require less attention. As it played out in the study region, each community needed to eventually move forward into other phases of the DFC process, while still tending to phase 1.

#### 3.1.2. Phase Two: Engage

Dementia Friendly America has identified two key activities to accomplish during the second phase of dementia-friendly community building: (1) Engage key or broad community membership in a dialogue to learn about community strengths, gaps, and priorities for action; and (2) Identify ways that the community can build on its assets and fill gaps to provide additional support. Both activities assume that a community has a baseline knowledge about an issue like dementia, its impact on the community, and the services and support that can be mobilized to strengthen the community environment for PLWD and their care partners. This meant that communities had to build some level of community readiness before they could engage in these discussions and that communities that joined the program in the later years of the initiative had less time to accomplish this work. The communities that more fully implemented phase 2 activities were those engaged in the program for two years or longer and that met one or both of the following: coordinators had focused on readiness building among the Action Team and/or Action Teams were comprised of four or more community sectors. Appendix A provides the results from the individual scoring of the communities on each of the four activities.

For all the regional DFCs, the focus of the phase 2 work was often internal to the Action Team. Action Team members were often recruited to represent sectors that would have information about available resources and gaps in services for PLWD and caregivers. Action Team meetings were then used to dialogue and share this information with other community sectors represented on the team. In the four communities that did launch more formal local needs assessments, Action Team members recruited additional stakeholders from respective sectors to participate in the DFA-adapted sector questionnaire and the BKAD, administered online by the university.

#### 3.1.3. Phase Three: Analyze

Phase 3 of the DFC change process involves two core activities: compiling and interpreting the data gathered during phase 2 and using the compiled data to draw conclusions and select priority goals that the community might undertake. Only four of the regional DFC communities embarked on formal, local needs assessment activities using the BKAD or the DFA tool (see Table 3). Of the four communities that gathered needs assessment data, only two yielded sufficient data from both tools to inform local assessment efforts. One community administered only the BKAD, and another community gathered only sufficient sample sizes through the wider regional assessment that was conducted.

**Table 3 ijerph-22-00721-t003:** Number of community sectors and individuals participating in DFC assessments, June 2024.

Cohort and Community Identifier	DFA Community Engagement Tool		BKAD	
	Sectors	Participants	Sectors	Participants
1A	3	9	7	27
1B	4	14	8	35
2A	1	1	6	21
2B	3	18	5	12
3A	0	0	0	0
3B	0	0	0	0
4A	0	0	0	0
4B	0	0	0	0

Note: The number of possible sectors totaled nine as all healthcare subsectors were combined.

Almost all the communities recruited community residents to participate in a regional assessment conducted by the university. The DFCs that more fully implemented phase 3 were the communities that had the longest standing Action Teams and involvement in the GWEP DFC program. The other four DFCs had not begun assessment activities at the time of the grant award’s conclusion. Even the early cohorts did not prioritize analyzing the results and using these results to inform programmatic planning. Prior to concluding the funded DFC program, the university compiled results for each community that had gathered local data into a slide deck to be shared and discussed with the Action Team. Refer to Appendix A for information on the variation between communities in implementing phase 3.

#### 3.1.4. Phase Four: Act

Phase 4, Act, is the phase where DFCs share the results of the assessment with the broader community, gather input on priority goal areas, develop an action plan that prioritizes opportunities and goals, and seek funding where needed. All the DFCs used action teams to develop and implement informal, annual action plans. Community action plans, however, were not developed with the formal format of a strategic plan nor with the intention of sharing the plan with the community, as the DFA-defined phase four suggests is needed. Instead, the DFC action plans focused on the 2–3 events that would be offered in the community that year and any additional services (e.g., physician office dementia gift bags that included fidget items, activity backpacks to be checked out from the library by caregivers and persons living with dementia), or action team activities.

In summary, regional DFCs demonstrated a greater adherence to the first two phases of the DFC change model than the subsequent phases. Figure 1 illustrates this pattern, while also showing community-level variation in adherence.

The evaluation also observed that the only adherence to phase 3 was an artifact of the university’s work. To date, the university staff do not know if needs assessment data, local or regional, have been shared with the Action Teams following the grant’s conclusion to inform planning.

### 3.2. Adherence to DFC Guiding Principles

In addition to assessing community adherence to the DFC process, the evaluation examined the extent to which participating communities demonstrated alignment to the two DFC guiding principles defined by DFA on its website and in the Dementia Friendly Community Evaluation Guide. The first of these principles focuses on the engagement of PLWD in the four phases of DFC building to ensure that the needs and wishes of PLWD are centered in the community initiative. The second focuses on the collaborative leadership required to build dementia-friendly communities. The evaluation examined monthly reports, action team rosters (see Appendix A), and meeting notes to determine whether these guiding principles were evident. Then, the evaluator validated the information with the university GWEP program coordinator prior to sharing the results with the DFC coordinators and action teams for final validation and discussion.

#### 3.2.1. Principle One

Given the significant investment of DFCs in building Action Teams, the analysts first examined the team rosters to determine if the first principle was present and what might be learned. With the exception of the first cohort, regional DFCs were not successful in recruiting PLWD to the action team by the grant′s close. Table 4 details the representation of PLWD by DFC and offers a comparison with a different interest group, caregivers. As shown, DFCs were more successful in recruiting caregivers than PLWD to the action team. The same two DFCs that were successful in recruiting PLWD additionally recruited caregivers to the action team; DFCs from the second and fourth cohorts, however, also were successful in recruiting caregivers.

The monthly reporting and Action Team notes indicated that PLWD recruitment was a standing agenda item for most DFCs. The university GWEP DFC program coordinator reported that DFCs perceived the stigma of dementia to be a significant barrier to recruitment. The university GWEP program coordinator also shared several strategies that communities had used to help compensate for the lack of active PLWD on action teams, including gathering caregiver–PLWD dyads on an ad hoc basis or as an advisory group.

#### 3.2.2. Principle Two

The second DFC principle focuses on the need to collaborate across sectors to transform community environments into dementia-friendly ones. The evaluation did not assess how effectively the action teams worked together, but we did observe that all the teams met regularly (monthly or bimonthly) throughout the program period. The evaluation also examined whether the Action Teams reflected multiple sectors in the community and if the size of the action team meeting in the final year of the grant suggested the continuation of the group.

The DFA community change framework expects action teams to be multi-sectoral to comprehensively address the needs of PLWD and their caregivers within the community. DFA defined nine community sectors: business, caregiver services and support, community members, community services and supports, the faith community, legal and financial, local government, residential settings, and healthcare. In addition, DFA defined four sub-sectors of healthcare (clinics, home care, hospitals, and nursing homes), one of which, nursing homes, DFA notes, overlaps with the residential setting sector. Table 5 reports on the number of action team members and the number of sectors by DFC.

In the final year of the GWEP DFC initiative, 91 community members served on regularly meeting action teams. The size of the individual action teams grew and ebbed over time, with most communities having action team memberships of 11 or more by the end of the grant period that represented between four and six different community sectors.

Although action teams survived multiple years, the monthly reports, action team meeting notes, and validation with the program coordinator indicated that there was change in the Action Team composition and leadership over time. There were action team members who left one DFC to join a newly established one that was closer to their home or work. There also were action team members who participated in multiple action teams, for example, if multiple communities fell within the service area of their primary place of employment. Action team members also moved to a new place of employment or residence, impacting action team participation. There also were some members whose participation was simply not sustained over time. The evaluator, however, observed that all the DFCs continued to recruit new members post DFA recognition, and that recruitment was informed by the new sectors that the action teams were attempting to engage in the community effort.

During the initiative, there also was a transition in action team leadership in four of the eight communities. In three of the four communities, this transition was temporary but required capacity building among action team members to take on leadership functions. Despite changes in the action team membership and leadership, all the DFCs indicated that they would continue local initiatives after the grant concluded and asked that the university continue to convene them at a regional level. In addition, two of the DFCs that started as city initiatives had expanded to the larger county, and two additional sub-county DFCs (communities 2B and 3A) had consolidated efforts to focus county-wide.

### 3.3. Alignment with National DFC Efforts

To assess the alignment of regional DFC activities with other DFCs working across the nation, the evaluation examined the fit between the six emerging goals of DFCs nationally: (1) increase awareness and understanding of dementia and of PLWD; (2) increase awareness and understanding of brain health and risk reduction; (3) collaborate with public, private, nonprofit, and health care sectors to better serve PLWD and care partners; (4) address the changing needs of people with dementia and care partners; (5) create a supportive social, cultural, and business environment that is inclusive of those living with dementia; and (6) improve the physical environment in public places and systems to be dementia-friendly [47]. The Dementia Friendly Community Evaluation Guide provides several example activities for each goal. These activity lists were used to locate each regional DFC community activity with the best fitting goal. See Appendix A for the categorization of the DFC activities by national DFC goal.

Although GWEP DFCs did not utilize community data to inform planning, their activities did align with the six goals that DFA has identified as common to DFCs nationally. Communities were most likely to address one of the two following goals:Increase awareness and understanding of dementia and PLWDAddress the changing needs of PLWD and care partners.

As Appendix A shows, 13 of the 48 community events reported to the university focused on increasing awareness and understanding. All the communities reported at least one activity with this aim, often involving a local chapter of the Alzheimer’s Association in community-based education delivery. Another 23 community events offered by six of the eight communities focused on addressing the needs of PLWD and care partners. These activities ranged from care baskets or education for caregivers to Memory Cafe-style events that offer caregivers and PLWD a space to make new memories together in a public space without concerns about fitting in with others.

There were no activities that were categorized as collaboration across sectors to better serve PLWD and care partners, as the evaluation guide indicated that these activities should result in policy or system-level changes in how PLWD are served. DFCs in our region, however, did collaborate across sectors to organize caregiver resource fairs that supported care partner knowledge of available resources. For most communities using this strategy, only one fair had been offered to date. These fairs were categorized under the goal of addressing the changing needs of PLWD and care partners, as these fairs aimed to equip community members for the different stages of disease progression, rather than initiate community level change in how sectors collaborate.

The other two areas where regional DFCs had not focused their work yet were:Increase awareness and understanding of brain health and risk reductionImprove the physical environment in public places.

Although three communities did address the first of these two goals, the leading contributor was the university, community 2B, which held four of the six community events that were coded to this goal. Only two other communities had worked in this area prior to the end of the grant, and each had offered only a single event. Even fewer communities had worked on the second goal of improving physical environments in public places and systems to be dementia-friendly. The one community that had worked in this area had integrated dementia training for new firefighter and paramedic orientation to improve public safety for PLWD. Communities were more likely to address the social and business environment to ensure that it was inclusive of PLWD.

This information was shared with DFCs at the grant-end celebration to highlight where DFCs had focused their work as a region and the future goal areas that might be prioritized in the future. Increasing awareness and understanding of brain health and risk reduction was highlighted as an important goal of future work, as the regional needs assessment found that awareness and knowledge of brain health and risk reduction were lacking.

## 4. Discussion

This project established a multi-year network of DFCs and carved out a role for continued academic–community partnership beyond grant funding. The evaluation examined community adherence to the four-phase process of dementia-friendly community building and alignment with the DFA guiding principles to assess the successes and identify potential areas to address through future DFC implementation. Overall, the study region demonstrated engagement with the guiding principles and each of the four phases. Further, the project indicated that universities can play an instrumental role in helping DFCs launch new phases and make progress collectively as a region in all phases. The evaluation also identified some areas for future DFC process elaboration, refinement, and additional research.

Despite prior dementia workforce development in the state and regionally, the midwestern university was uncertain of whether we would find regional commitment to building DFCs in the surrounding urban and rural counties and interest in joining our initiative. We were pleased to find that community stakeholders saw value in joining an academic–community partnership and a network of other communities to help accomplish dementia-friendly changes. The university contributed resources that were valuable to the community process (i.e., Dementia Live certifications and kits, activity kits, and books that could be made available to the community through public libraries, as well as waiting rooms in healthcare organizations, etc.). Helping community stakeholders to access these resources or simply serving as a financial pass-through for resources to reach the community was an important function of the university’s role. In addition, the university’s regular presence at action team meetings and other community events helped to support emerging leadership and elevate dementia as a community issue. In one case only, the university supported a DFC that likely would have launched without the university’s support. In most communities, the university played an important role in increasing community readiness for action. The university served as a central hub of information, resources, and guidance on DFC implementation, and helped communities to apply these tools. Moreover, as a central hub, the university was well positioned to monitor the training and planning needs of communities and could anticipate when virtual training or other opportunities would be helpful to multiple communities or the entire network. The university, at times, convened communities to learn from one another’s work and helped to establish mentoring relationships between longer established communities and newer ones. In the case of community needs assessments, when there were not sufficient technical resources to support individual communities in conducting the work, the university was able to devise methods for building on the locally completed needs assessment activity to conduct a region-wide needs assessment that would benefit all, albeit providing less localized information.

Walking alongside communities as they underwent the DFC change process also gave university staff insights into where further specification of DFA-defined community phases would be helpful. One role that we played was to identify gaps and obtain appropriate tools that could be adapted to support the community process. The university drew on other research-based information in the field of dementia, public health, and community planning to address these gaps. Some examples of the gaps and challenges we encountered are discussed below.

In working with DFC coordinators, university staff identified several aspects of phases 1 and 2 that were not sufficiently explained to support implementation, including how to assess and address community readiness and how to identify stakeholders to inform community assessment activities. Despite decades of research that indicates that community change must be approached differently based on the level of community readiness or preparedness to act [36,49], the DFA framework does not point communities to examples of community readiness tools that they might use or address how the four-phase process of building a dementia-friendly community might look different depending on this readiness.

The Community Readiness Theoretical Model is an established theory of community change that acknowledges that communities must be ready for specific change and that readiness is measurable [36]. This validated model has been effective in sustained community change across broad areas of prevention [50], health education [36], and other public education efforts [36,50,51]. This model assesses the stages of readiness with the goal of accurately defining the stage of community change readiness. This process ensures that the right level of intervention is implemented at the right time and in the right way [50]. Since community readiness is an important aspect of sustainable community change, incorporating existing tools and theory, such as this one from the community prevention field, would strengthen the DFC framework and support communities in trying to effect change.

Further, although the DFA-defined community change process is conceptually recognizable from public health and social service planning models that emphasize the importance of stakeholder identification, the establishment of multi-sectoral leadership to drive, and the use of community data and assessment activities to inform the contents of local plans for change, DFA does not offer this contextualization. Thus, communities may not be learning to align themselves with others addressing chronic disease within the community or how to adapt existing community planning tools and guidance to the DFC process. There is a wealth of existing resources in communities, regions, and states already used to support community planning (see an overview of the models, intended audiences, and who promotes use at the Centers for Disease Control and Prevention website) [52]. Bringing public and nonprofit sectors on board will eventually expose DFC action teams to these frameworks and models, but it could be very helpful for academic partners to introduce action team members to them upfront and translate their utility to the DFC process.

Greater specificity about what it looks like to implement the four phases effectively is needed. For example, how should involvement of PLWD throughout the four-phase process be achieved? Is it appropriate, or at least a good beginning, for communities like those that we worked with, and those surveyed by Mathie (2022), to engage PLWD in shorter-term opportunities for community input [28]? Does the involvement of caregivers help to achieve the goal of ensuring that the initiative has meaningful results for PLWD? As DFA continues its leadership in community-based change for PLWD and their families, we anticipate that communities will increasingly look to DFA for guidance on these topics.

Working towards the development of DFC “implementation fidelity tools” or “checklists” would help to advance this international movement by clarifying how the intervention is intended to be implemented and, thereby, helping to promote the success of its translation into practice [53]. Implementation tools or checklists would lay out expectations of what DFC implementation involves, and identify benchmarks that can be used to self-assess and improve efforts locally. While the DFA’s *Community Evaluation Guide* is a very helpful tool to communities, it primarily focuses on how to assess the programs and services that DFCs implement after completing phase 4. Recommended tools would guide and support the evaluation of the community process itself, its alignment with DFC principles, and the sufficiency of community activities for accomplishing phases. There are larger picture questions as well about the framework for DFC change that would be helpful for future attention. For example, when is a community ready to move on to the next phase? Is some reiteration of the phases normal as new sectors join the community process? And, when and how is it appropriate to adapt the framework? It also might be helpful to assign time expectations to the different phases. In our region, where the readiness for multi-sector action appeared relatively low initially, action teams spent several years building representation across sectors and still had not addressed all the sectors that the DFA defines. Tool development might be accomplished through academic or other technical assistance partnerships at the national and/or local levels.

A final area where DFA might consider providing more guidance is where to find sustainable leadership for DFCs. Over time, and as we worked with several cohorts of communities, the university became more aware of the latent leadership in long-term care facilities and aging services, and was well poised to coordinate the local initiative. This may be related to other environmental factors in this state, as Area Agencies on Aging were aligning themselves with Dementia Friends, a training initiative of Dementia Friendly America, which helped to meet the training needs of Action Teams and offered a valuable scaffold for sustaining elements of the program beyond GWEP grant funding. However, the recent federal emphasis on the issue of dementia suggests that the environment nationally is ripe. Helping communities to identify leaders who can help to integrate DFC work into their professional roles may lend a greater stability to DFCs than, for example, being volunteer led.

The DFCs established as part of the GWEP program were equally likely to be established in geographic areas that were designated as medically underserved areas (MUAs) as in those that were not. However, DFCs were somewhat more likely to be rural communities. The researchers do not know the extent to which the DFC process or the approaches taken by the university were more conducive to rural areas. However, there are many examples of planning processes and tools that have been successfully used within more urban county planning projects. Given that the program targeted rural counties, we have less information on the implementation of the approach in urban areas. Suburban and urban implementations often identify a lack of community connection as a significant barrier to success [21]; however, they often have access to more resources.

## 5. Conclusions

This descriptive project highlights the value of an academic–community partnership in establishing a multi-year network of DFCs. The project results suggest that academic partners can support communities in determining how best to implement the steps involved in building a DFC, and perhaps, at a national level, help to add further clarification on the framework that will advance the DFC movement overall. The establishment of academic–community-based partnerships in creating dementia-friendly communities is critical because the current lifetime risk of dementia is higher than once thought and communities are increasingly going to be challenged to provide support and environments where all can thrive [54]. The project team identified opportunities for academic partners to not only support efforts within the DFC network, but to expand the potential impact at the regional and state levels.

Universities can contribute locally and nationally to the DFA four-step model by filling in adherence ambiguity or gaps, and by identifying tools to be brought in. Universities can identify whether the tools that are already being used in community partnerships for other purposes can be adapted effectively for DFC initiatives. Universities, working alongside the DFA and local communities, can also provide expertise and research capabilities to define and establish implementation and adherence protocols. For example, universities can assist communities in developing protocols to adhere to this step. Another reciprocal benefit that universities can support communities with is building networks to increase communication and learning. Universities are in a position to work with multiple communities to support and guide action teams to learn from each other and share motivating experiences. In our project, the result of these networks not only benefited communities but also increased the knowledge and understanding of university–community partnerships; communities learn from the efforts of the university and the university learns from the interaction with the community. Our reciprocal partnerships worked well in assisting in the needed supportive avenues of connection and collaboration, as evidenced in new communities continuing to reach out to join our network of DFCs.

University faculty and staff involved in academic–community dementia-related partnerships have opportunities to optimize efforts by integrating public health models to improve brain health across the lifespan, increase knowledge about dementia, and reduce stigma surrounding brain change. As an example, university faculty and staff may consider taking leadership roles in educating governing bodies about frameworks, such as the DFC community change process, on Dementia State Plans. Initiated and supported by state governments, Dementia State Plans are comprehensive data-informed documents that are intended for a widespread impact available in most states. Likewise, most states are engaged in Multisector Plans on Aging (MPAs), which are 10+ year blueprints developed by cross-sector stakeholders addressing the needs of older adults. Integrating academic and community partnerships to promote dementia-friendly initiatives in MPAs has the potential to build a state-level capacity to respond to the expected growth in the prevalence of dementia.

Moving forward, universities can play a growing role in the establishment of DFC networks. The DFC work is unique in the comprehensive change it asks communities to undergo to become dementia-friendly. In the spirit of cross-sector collaboration, future research can examine the impact of this change at the community, region, and statewide levels, when universities partner with DFA to support DFC networks.

## Figures and Tables

**Figure 1 ijerph-22-00721-f001:**
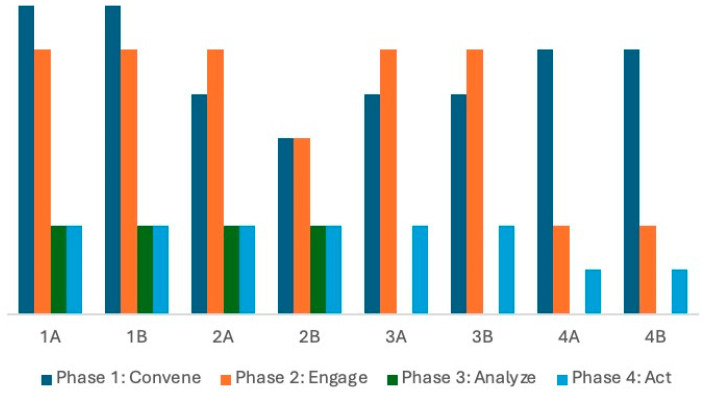
Community adherence to the four phases of the DFA model.

**Table 1 ijerph-22-00721-t001:** DFCs by date of designation and geographic descriptors.

Cohort and Community Identifier	Date of DFC Designation	Population Density *	Medical Designation *	Original Geographic Scope	Sector Represented by the DFC Coordinator **
1A	January 2021	Rural	MUA	Town	AAA
1B	January 2021	Rural	MUA	Town	AAA
2A	February 2022	Rural	MUA	Town	LTC
2B	December 2021	Urban	Not MUA	Neighborhood	University
3A	October 2022	Urban	Mixed	Neighborhood	LTC
3B	December 2022	Urban	Not MUA	County	LTC
4A	November 2023	Rural	Not MUA	County	LTC
4B	May 2024	Mixed	Mixed	County	LTC

* The research team used the Health Services and Resource Administration’s Rural Health Grants Eligibility Analyzer and Medically Underserved Area (MUA) Finder to determine the population classifications. ** AAA = Area Agency on Aging; LTC = Long-term Care.

**Table 2 ijerph-22-00721-t002:** DFC activities and university technical support.

DFA Model Step	Key GWEP DFC Activities	University Technical Support
Step 1: Convene key community leaders and members, known as an action team, to understand dementia effects	Action Team formation Community applications for DFA recognition Regular Action Team convenings	Recruited and trained DFC facilitators Partnerships with two regional Area Agency on Aging organizations to facilitate stakeholder informational meetings in communities DFC overview slide deck adapted for each community included a “demographics of dementia in your community” slide based on the DFA-provided slide deck Demographics tool template Action team roster templates Meeting summaries
Step 2: Engage key leaders to conduct a community assessment of the factors supportive of PLWD and those that are barriers	Surveyed community leaders and members using the Basic Knowledge of Alzheimer’s Disease (BKAD) * Surveyed sector representatives using the Community Engagement Tool, an assessment tool in the DFA Community Toolkit **	Researched measures to inform regional and local community education Developed an Institutional Review Board proposal to study and share results of the BKAD and DFA Conducted a regional assessment using two assessment tools, the BKAD and DFA’s sector assessment tools from the Community Toolbox
Step 3: Analyze assessment results to determine stakeholder issues and set community goals	Action Teams discussed assessment tool questions and ideas that these prompted for programming Action Teams supported regional assessment activities	Community assessment results slide deck prepared for Action Teams Zoom-based reviews of assessment results and slide decks with facilitators Hosted in-person celebration with DFC Action Team representatives to share regional assessment and evaluation results Network celebration engaged DFCs in brainstorming how the university could sustain program support
Step 4: Create a community action plan that includes specific objectives, activities, leadership, and timelines	Regular Action Team meetings to plan quarterly or semiannual community events Community events held, as well as ongoing activities Participated in DFA network events (see next column)	Created and supported DFC network programming, such as an annual educational conference, Let’s Talk about Dementia; November Family Caregiver educational workshops; a traveling summit with Teepa Snow, dementia care expert; coordination with the Alzheimer’s Association state chapters to provide educational workshops and volunteer training to DFCs and supported Dementia Live^®^ coaching certifications and gear kits Participated regularly in the DFA state affiliate and shared national and state resources with local DFCs Supported local DFCs in organizing events that aligned with national dementia initiatives (e.g., Take It to the Streets) Web-based DFC Corner was created to house public educational resources and sample items created by the network

* BKAD—the Basic Knowledge About Alzheimer’s Disease (BKAD) [46] (see Table 3 for participation), designed specifically for rural communities to assess knowledge about the disease. ** The Community Engagement Tool Instructions [31], (see Table 3 for participation), also available for download from the DFA website, offered a set of questions for each of nine DFA-defined sectors, and four additional sub-sectors of healthcare.

**Table 4 ijerph-22-00721-t004:** PLWD and caregiver Action Team membership, June 2024.

DFC	PLWD Representation	Caregiver Representation
1A	1	1
1B	1	2
2A	0	0
2B	0	1
3A	0	0
3B	0	0
4A	0	0
4B	0	3

**Table 5 ijerph-22-00721-t005:** June 2024 snapshot of GWEP DFC Action Teams.

Community Identifier	Number of Action Team Members	Number of Sectors Represented	Geographic Scope
1A	17	6	County
1B	11	6	County
2A	11	4	City
2B	6	2	(Merged with 3A)
3A	11	5	County
3B	13	5	County
4A	14	4	County
4B	8	3	County

## Data Availability

The datasets presented in this article cannot be de-identified for public use. Inquiries requesting data information or data extracts can be directed to Dr. Laurel Standiford Reyes.

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
