# Peer review of "A University’s Role in Developing a Regional Network of Dementia Friendly Communities"

_ijerph, 2025, doi:10.3390/ijerph22050721_

Round 1
Reviewer 1 Report
Comments and Suggestions for Authors
I would like to congratulate the authors on their detailed description of the project they took part in. Although complex, you present the main objectives of your participation as a university, and present the results achieved. It's relevant, well-written and fulfils the requirements of an article. Methodologically, it is well explained.
However, I consider that some improvements can be made:
1- Starting with the abstract, I think you could improve it. A division into introduction, objectives, results, discussion and main conclusions would be a good approach to direct the reader to your article. The way you present it is too broad.
2- The introduction is too long, but at the end it clearly explains the aim of the project and the university´s role. It is understood that the division made in the introduction has served to direct the reader to the basic points of the creation of community support networks for families of people with dementia.
3- On the discussion section (line 553) there´s a reference without number (Mathie, 2022). Please update.
4- I would ask you to put a list of acronyms at the end of the article, as there are a lot of acronyms used, and this list can make your article easier to read and understand.
Reviewer 2 Report
Comments and Suggestions for Authors
This is, as the authors say, a descriptive paper that reports on and evaluates a community project organized in accordance with a national template. It has some value as a public record of the project and will be of some interest to those planning similar ones, but it makes a limited contribution to the research base. A more substantial contribution might entail the use of a theory of community change or contribute to the wider debate on activities appropriate to DFCs.
There is a couple of minor typos: in table 2:1 a change of font on line 2B; and on p8 line 268 some unnecessary word, to evaluate
Reviewer 3 Report
Comments and Suggestions for Authors
My Reviewer Notes:
In this article, the authors describe partnerships between community organizations and university teams. These partnerships were formed to create and sustain dementia friendly communities. Community-based applied research that focuses on the roles that universities play is valuable. Descriptions and analysis of community-university partnerships promote, as the authors state, “increasing community readiness for action”, a timely topic that needs to be disseminated. In the following, I provide feedback on the introduction, purpose statement, research procedures, methods, results, and discussion.
The introduction and literature overview are presented well.
The purpose statement is defined at the end of the project purpose description. Please, check that the first purpose statement is accurate. It seems vague compared to subsequent two different purpose statements. In the discussion section, the authors define the purpose more clearly with a clear focus on the evaluation of the adherence to the four-phase process. At the beginning of the conclusion, the authors point to the article as a descriptive project. I suggest reviewing the different statements and writing a more succinct statement at the beginning that will clearly align with the aims of the article and presented results.
Methods are not described adequately. Why are materials and methods under the heading and section? The authors primarily describe procedures rather than methods. As reader I need to know exactly what research methods the authors applied as they conducted the evaluation of the community partnerships.
Methods should be described in terms of data collection and analysis. Readers need to know how exactly the retrospective evaluation of the regional network was conducted. How were surveys developed, how were they administered? Were there other data collection methods used (e.g. interviews or focus groups)? This all needs to be discussed before the results are presented. I suggest creating two sections, one titled Procedures and the other Methods.
The study participants need to be discussed in terms of how many participated in the surveys or any other data collection methods. While the authors mention why there were no participants from four cohorts later on in the results section, the reader would benefit from knowing this earlier in the paper. What type of data was collected during dialogues and needs assessments? How did the researchers evaluate the engagement of the cohort communities throughout the four phases? How did they gather information about the activities each of the cohort communities implemented?
The results section includes descriptions of methods the researchers used to evaluate particular activities and phases. The authors should put the descriptions of data collection in the methods section. Then the results are more succinct and more clearly present the activities outlined in the tables.
For example, under 3.2 Adherence to DFC Building Principles discussed in the Results section, the first paragraph needs to be in the Methods section. “In addition to assessing community adherence to the DFC process, the evaluation examined the extent to which participating communities demonstrated alignment to the two DFC guiding principles defined by DFA and addressed on its website and in the Community Evaluation Guide. The first of these principles focuses on the engagement of PLWD in the four phases of DFC building to ensure that the needs and wishes of PLWD are centered in the community initiative. The second focuses on the collaborative leadership required to build dementia friendly communities. The evaluation examined action team rosters and meeting notes to determine whether these guiding principles were evident. Then, the evaluator validated the information with the university GWEP program coordinator prior to sharing results back with DFC coordinators and action teams for final validation and discussion.” In the methods section, describe exactly how the evaluation was conducted!
In the results section, examples of specific activities would help readers to paint a picture of individual communities’ approaches. Right now, the activities are listed very generally, like titles.
Clarity of writing is generally good. Checking that long run-on sentences are edited into at least two separate ones will help readers to better follow the authors’ thoughts.
For example:
Lines 47 to 53: Create two sentences here.
Lines 361 to 362: What are needs assessment result slide decks? How were they developed and used?
Lines 513 to 517: More clearly describe what the university team did to help communities conduct needs assessments.
Organization of thought is good in terms of headings and subheadings. Presentation can be improved by including text in the correct sections as suggested earlier in the review. The methods section needs to be rewritten to help readers understand exactly how the authors conducted the evaluation of the cohort communities and their adherence to the four-phase process.
All the best with the revision of the manuscript!
Round 2
Reviewer 3 Report
Comments and Suggestions for Authors
Thanks for sharing a thoughtfully completed revision. With the better organization of sections and content, I can follow the study procedures and results more easily. Applied community-based research is complex. With the revisions, I understand the complexities of the presented evaluation better and hope readers will learn valuable insights from the process that the university and community partners underwent.
Make sure to use consistent formatting of headings and subheadings. For example, under 3. Results. The first subheading 3.1. is bolded. The second one 3.2. and the third one 3.3 are not bolded. They need to be the same format.
Please, explain that MUA stands for Medically Underserved Area. Not all community leaders may know this designation.
Community-university partnerships and initiatives such as the one described by the authors need to be promoted. The authors are contributing to this vital area of scholarship.
